# A master regulator of central carbon metabolism directly activates virulence gene expression in attaching and effacing pathogens

**Kabo R. Wale**[1], **Nicky O'Boyle**[2,3,4], **Rebecca E. McHugh**[1], **Ester Serrano**[1], **David R. Mark**[1], **Gillian R. Douce**[1], **James P. R. Connolly**[5] *, **Andrew J. Roe**[1] *

**1** School of Infection and Immunity, University of Glasgow, Glasgow, United Kingdom, **2** School of Microbiology, University College Cork, Cork, Ireland, **3** Department of Pathology, School of Medicine, University College Cork, Cork, Ireland, **4** Department of Microbiology, School of Genetics & Microbiology, Moyne Institute of Preventive Medicine, Trinity College Dublin, Dublin, Ireland, **5** Newcastle University Biosciences Institute, Newcastle University, Newcastle-upon-Tyne, United Kingdom

* James.Connolly2@newcastle.ac.uk (JPRC); Andrew.Roe@glasgow.ac.uk (AJR)

**Data Availability Statement:** The RNA-seq data is available from the European Nucleotide Archive under the accession number PRJEB74273.

## Abstract

The ability of the attaching and effacing pathogens enterohaemorrhagic *Escherichia coli* (EHEC) and *Citrobacter rodentium* to overcome colonisation resistance is reliant on a type 3 secretion system used to intimately attach to the colonic epithelium. This crucial virulence factor is encoded on a pathogenicity island known as the Locus of Enterocyte Effacement (LEE) but its expression is regulated by several core-genome encoded transcription factors. Here, we unveil that the core transcription factor PdhR, traditionally known as a regulator of central metabolism in response to cellular pyruvate levels, is a key activator of the LEE. Through genetic and molecular analyses, we demonstrate that PdhR directly binds to a specific motif within the LEE master regulatory region, thus activating type 3 secretion directly and enhancing host cell adhesion. Deletion of *pdhR* in EHEC significantly impacted the transcription of hundreds of genes, with pathogenesis and protein secretion emerging as the most affected functional categories. Furthermore, *in vivo* studies using *C. rodentium*, a murine model for EHEC infection, revealed that PdhR is essential for effective host colonization and maximal LEE expression within the host. Our findings provide new insights into the complex regulatory networks governing bacterial pathogenesis. This research highlights the intricate relationship between virulence and metabolic processes in attaching and effacing pathogens, demonstrating how core transcriptional regulators can be co-opted to control virulence factor expression in tandem with the cell's essential metabolic circuitry.

## Author summary

This study reveals an unexpected role for a metabolic regulatory protein in the attaching and effacing family of pathogenic Gram-negatives. The transcription factor PdhR normally functions by repressing expression of the pyruvate dehydrogenase complex in

Publicly available genome sequences were obtained from the NCBI Sequence Read Archive. Each of these genomes cn be readily accessed from the accession numbers provided in S5 Table.

**Funding:** Research in the J.P.R.C. lab was supported by a Springboard Award from the Academy of Medical Sciences/Wellcome Trust [SBF005\1029], a Royal Society Research Grant [RGS\R2\202100], a Medical Research Council Career Development Award [MR/X007197/1] and a Newcastle University Faculty Fellowship. Research in the A.J.R. lab was supported by grants from the Biotechnology and Biological Sciences Research Council [BB/W015781/1, BB/V009494/1] and the Medical Research Council [MR/V011499/1]. K.R.W. was supported by an Eleanor Emery PhD Scholarship. The funders had no role in study design, data collection and analysis, decision to publish, or preparation of the manuscript.

**Competing interests:** The authors have declared that no competing interests exist.

response to cellular pyruvate levels, thus helping cells to manage their energy production. However, we discovered that PdhR is also capable of controlling the ability of these pathogens to colonise host cells early in infection. Deletion of *pdhR* affected the transcription of hundreds of genes, including many involved in the infection process. As such, PdhR was found to be required for host cell attachment both in vitro and in vivo. This level of control is achieved by a mechanism involving direct interaction between PdhR and the gene region encoding a key pathogen virulence factor. This discovery highlights how bacteria can re-purpose proteins involved in basic processes of life to control their ability to cause disease, showing that the connection between a bacterial metabolism and pathogenesis is more complex than previously thought.

## Introduction

To successfully establish an infection within their preferred host niche, bacterial pathogens have evolved diverse strategies to outcompete the resident microbiota and overcome colonization resistance [1,2]. Such strategies often revolve around the deployment of virulence factors that aid pathogens by granting them a competitive edge over species with similar nutritional requirements [3,4]. Cellular adhesion is one key functionality of a diverse range of virulence factors that includes fimbriae and large multi-protein complexes, such as the Type 3 Secretion System (T3SS) [5]. However, these systems are energetically expensive and therefore require carefully orchestrated regulation to ensure deployment under the most advantageous of scenarios [6]. Naturally these regulatory events will occur in response to changing environments as pathogens traverse the host environment and adapt their metabolic behaviour. Therefore, the regulation of virulence traits is dynamically modulated in response to cues from the host milieu, thus maximizing the pathogen's competitiveness during host-microbe interactions.

Enterohaemorrhagic *Escherichia coli* (EHEC) is a zoonotic pathogen that asymptomatically colonizes ruminant hosts [5,7]. Contamination of the food chain can therefore lead to human infection by EHEC, which causes severe diarrheal disease and can potentially lead to life-threatening haemolytic uremic syndrome. EHEC belongs to the attaching and effacing (A/E) family of pathogens [8,9]. A/E pathogenesis is mediated by a T3SS encoded within a pathogenicity island called the Locus of Enterocyte Effacement (LEE), which facilitates intimate adherence of EHEC cells to the apical surface of the colonic epithelium and the formation of pedestal-like lesions upon which the bacterium resides [10–12]. The LEE consists of 42 open reading frames over five polycistronic operons (termed LEE1 to LEE5) that collectively encode all the necessary components and primary effectors of a functional T3SS [13]. The T3SS acts as a molecular syringe, injecting over 30 effector proteins into host cells, which collectively subvert the host cell's normal function [14–16]. *Citrobacter rodentium* is an exclusively murine pathogen that also encodes the LEE. Like EHEC, *C. rodentium* relies on its T3SS to colonise the host. However, because it is a natural murine pathogen and able to outcompete the native intestinal microbiota, establishment of infection does not require antibiotic pre-treatment of the host. This has led to *C. rodentium* being adopted as the relevant surrogate model system supporting delineation of the mechanisms underlying A/E pathogenesis in vivo [17–19].

A multitude of host and microbiota-derived cues can be perceived by EHEC to signal regulation of the LEE in response to the environment. This includes nutrients and small metabolites such as sugars, amino acids, fatty acids, quorum sensing molecules and hormones [20–22]. These signals often have cognate transcription factors (TFs) or two component sensing (TCS) systems that respond to the respective cue by either promoting or repressing

transcription of target genes [23]. The master regulatory region of the LEE (an extended promoter region upstream of the LEE1 operon) acts as a "hub" for receiving such converging signals, containing binding sites for numerous core-encoded TFs that directly regulate this foreign pathogenicity island. This field has been extensively reviewed, but some recently described examples include interaction of ArgR, FadR, DcuR and ExuR which directly regulate the LEE by responding to L-arginine, long chain fatty acids, L-malate and galacturonic acid, respectively [20–22,24–27].

In addition to these highly defined regulatory pathways, there are also examples of TFs that directly regulate the LEE but do not have known small molecule ligands. For example, the highly conserved LysR type transcriptional regulator YhaJ is found in nearly all *E. coli* strains and despite not having a defined regulatory function has been shown to directly regulate LEE expression [28,29]. Similarly, there are many examples of environmental cues that modulate LEE expression from a relevant ecological context without a known cognate TF partner. For example, D-serine is a toxic metabolite that is abundant in urine and restricts EHEC to the gut niche by activating and SOS-like response and blocking T3SS expression [30,31]. Conversely, L-arabinose is one of the most abundant nutrients found in the colon (the major site of EHEC colonisation) and enhances LEE expression via its metabolism and generation of endogenous pyruvate [32]. Exogenous pyruvate has also been shown in other studies to enhance LEE expression suggesting that this important intermediate metabolite in central carbon metabolism acts as a key regulatory effector in EHEC [33]. However, the precise sensing mechanisms of these example molecules remain unknown.

In this study, we discovered that the TF PdhR is an important regulator of the LEE in both EHEC and *C. rodentium*. Canonically, PdhR senses cellular levels of pyruvate and acts as a master regulator controlling genes involved in central carbon metabolism in bacteria [34]. It exerts this control by directly repressing the *pdh* operon, which encodes the essential pyruvate dehydrogenase complex. This multi-enzyme complex catalyzes the oxidative decarboxylation of pyruvate to acetyl-CoA, a key entry point into the TCA cycle [35,36]. However, in EHEC deletion of *pdhR* affected transcription of hundreds of genes, with pathogenesis and protein secretion being defined as the two most strongly associated functional categories. We demonstrate that PdhR directly binds to a discrete motif within the LEE master regulatory region, thereby activating expression of the T3SS, and that its action at this site is not dependent on the uptake of excess pyruvate from the environment. Finally, it was revealed that PdhR is required for both effective colonization of the murine host by *C. rodentium* and maximal expression of the LEE in vivo. Thus, this work highlights that the core metabolic TF PdhR helps underpin the intimate relationship between virulence and metabolic processes in both EHEC and *C. rodentium*.

## Results

### The PdhR transcription factor contributes to LEE regulation in EHEC

We previously determined the transcriptome of *C. rodentium* in vivo during colonisation of its natural murine host [37]. This analysis revealed >130 differentially expressed genes (DEGs), many of which were involved in host-specific metabolism (such as 1,2-propanediol and L-arabinose) that intersects with virulence regulation. However, determining the direct regulatory mechanisms of such responses is challenging. To identify potential regulators of virulence in EHEC, we mined this dataset and shortlisted all *C. rodentium* TFs that were differential expressed in vivo. We further refined this shortlist to TFs that were conserved in EHEC, resulting in four candidates–*bssS*, *yfeC*, *rcnR* and *pdhR*. To determine if these regulators may play a role in virulence, we generated isogenic mutants for each TF in the EHEC strain TUV93-0.

We then screened these mutants using a LEE1:GFP reporter plasmid (GFP fused to the LEE master regulatory promoter region) as a proxy for T3SS transcription (Fig 1). This experiment revealed that deletion of *bssS*, *yfeC* or *rcnR* had no negative effect on LEE transcription but loss of *pdhR* resulted in a significant ($P < 0.01$) downregulation of the T3SS. Given that PdhR is responsive to cellular pyruvate and that pyruvate has been previously shown to enhance the T3SS, we next tested whether pyruvate supplementation altered this phenotype [32–34]. Supplementation of MEM-HEPES with 1 mM pyruvate significantly increased T3SS expression as expected but only in the wild type background. Conversely, LEE expression levels were equally attenuated in the Δ*pdhR* strain regardless of whether excess pyruvate was provided to the growth medium (S1 Fig). This result suggests two things–that pyruvate regulation of the LEE occurs via PdhR and that PdhR can regulate the LEE independently of sensing exogenous pyruvate from the environment.

## PdhR activates EHEC T3SS expression and promotes host-cell adhesion

Deletion of *pdhR* resulted in a growth defect when EHEC was cultured in MEM-HEPES, a phenotype that could be complemented by expression of PdhR in trans (Figs 2A and S2). This was not surprising considering its role in acting as a master regulator of central carbon metabolism. However, we next wanted to address whether the reduction in T3SS expression was a consequence of this fitness defect or a true regulatory event on the LEE. To test this, we repeated our assays in MEM-HEPES supplemented with 0.2% succinate, as succinate has previously been reported to counteract the negative effect of a *pdhR* deletion in *E. coli* K-12 [38]. Supplementation of the media eliminated the fitness defect observed allowing us to separate the effects of deleting *pdhR* on T3SS expression from growth rate related phenotypes. Strikingly, T3SS expression remained significantly attenuated ($P < 0.01$) in the Δ*pdhR* background under these conditions, suggesting that the effect of this TF on the LEE does not result because of any associated fitness defects (Figs 2B and S2). To address if this transcriptional event resulted in reduced T3SS function, we profiled proteins secreted into the surrounding medium by SDS-PAGE. Levels of T3SS-secreted effector portions Tir, EspB/D, EspA and NleA (as

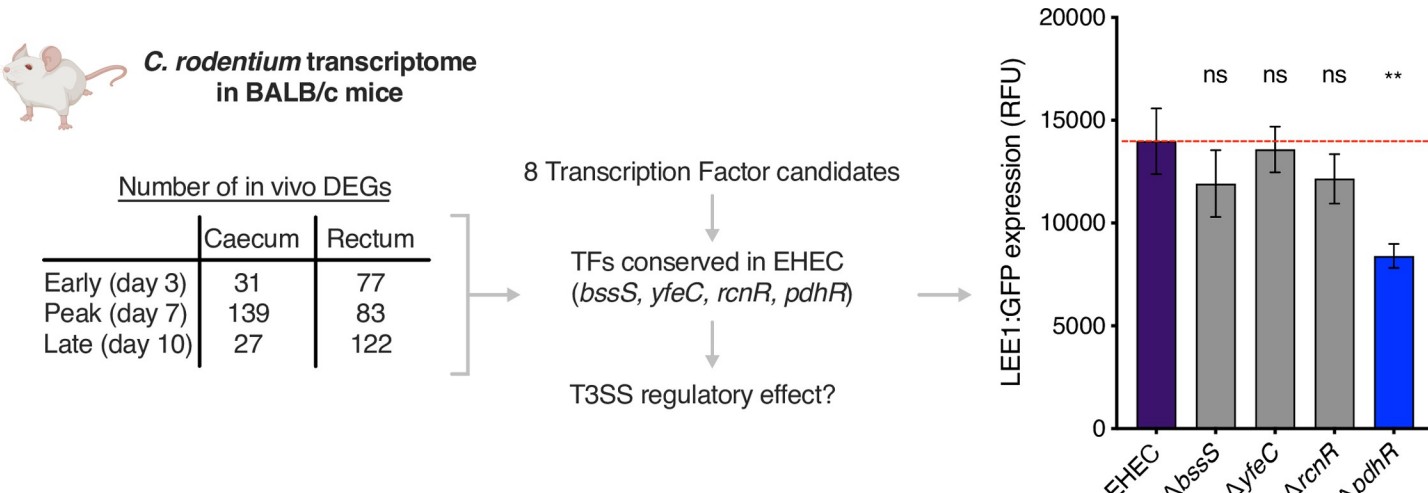

**Fig 1. Identification of PdhR as a regulator of the LEE-encoded T3SS in EHEC.** The flow chart summarises the *C. rodentium* in vivo transcriptome and illustrates the identification of four TF encoding genes in EHEC that were also found as part of the *C. rodentium* in vivo DEG set. The bar chart shows the results of a transcriptional reporter screen (using plasmid pLEE-GFP) to identify the effects of TF deletion on T3SS expression. ** and ns indicate $P < 0.01$ or not significant respectively, as determined by two-way ANOVA with Dunnett's post-test. Error bars represent standard error of the mean. Figure created using Biorender.com.

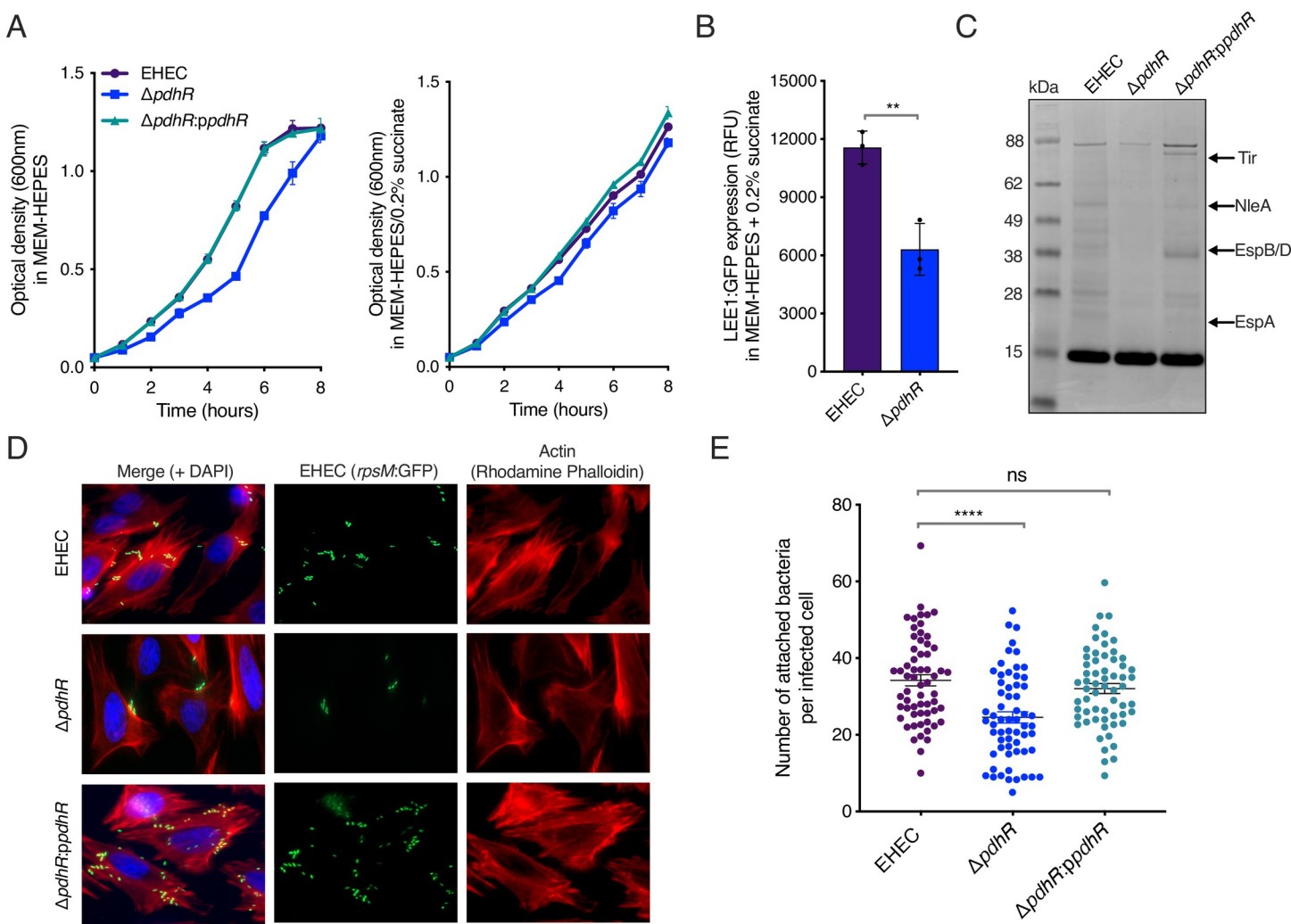

**Fig 2. PdhR promotes T3SS expression and host cell attachment independently of its effects on growth.** (A) Growth curves depicting hourly optical density (600 nm) measurements of EHEC, Δ*pdhR* and Δ*pdhR*:p*pdhR* cultured in MEM-HEPES (left panel) or MEM-HEPES supplemented with 0.2% succinate (right panel). Error bars represent standard error of the mean. (B) LEE-GFP reporter assay of EHEC and Δ*pdhR* cultured in MEM-HEPES supplemented with 0.2% succinate. ** indicates *P* < 0.01, as determined by two-way ANOVA with Dunnett's post-test. Error bars represent standard error of the mean. (C) SDS-PAGE profiling of EHEC, Δ*pdhR* and Δ*pdhR*:p*pdhR* culture supernatants from MEM-HEPES supplemented with 0.2% succinate. The arrows indicate the position of known and previously verified T3SS-related effector proteins. The large band at 15 kDa represents Lysozyme added equally to each sample as a loading control. This experiment was performed on three independent occasions. (D) HeLa cells infected with EHEC, Δ*pdhR* and Δ*pdhR*:p*pdhR* that were pre-cultured in MEM-HEPES supplemented with 0.2% succinate. The overlaid channels are labelled across the top. The images represent a single field of view from *N* = 10 for each strain. This experiment was performed on three independent occasions. (E) Quantification of data derived from widefield fluorescence microscopy analysis of HeLa cell infection assays depicted in panel D. **** or ns indicate *P* < 0.0001 or not significant respectively, as determined by two-way ANOVA with Dunnett's post-test. Error bars represent standard error of the mean.

previously determined by this method) were visibly detectable in the EHEC culture media but dramatically reduced in the supernatant of Δ*pdhR* cells (Fig 2C). Importantly, levels of secreted effectors were restored when Δ*pdhR* was complemented in trans. Lastly, we tested whether this reduction in T3SS expression and function would result in a decreased ability to colonise host cells. HeLa cells were incubated with equal numbers of wild type EHEC, Δ*pdhR* and the complemented mutant (Δ*pdhR*:p*pdhR*) and A/E lesion formation was assessed by fluorescence microscopy. Note that bacterial cells were transformed with an *rpsM*:GFP reporter plasmid whereas host cell actin was stained in red, allowing us to visualise co-localisation of bacterial cells with areas of condensed actin indicating the formation of an A/E lesion (Fig 2D).

Quantification of these adhesion events revealed that Δ*pdhR* formed significantly ($P < 0.0001$) fewer lesions per infected HeLa cell, confirming that downregulation of the LEE in the Δ*pdhR* background results in reduced T3SS activity at the cellular level (Fig 2E). Taken together, these results identified an important role for the metabolic regulator PdhR in controlling EHEC virulence gene expression and activity.

## PdhR is a global regulator of transcription in EHEC affecting several infection-relevant loci

The data thus far identified PdhR as a critical regulator of virulence gene expression in EHEC, suggesting pathogen-specific adaptations within the cell's regulatory network. To investigate the full extent of PdhR activity in EHEC, we performed RNA-seq transcriptome analyses of wild type and Δ*pdhR* cells cultured in MEM-HEPES supplemented with 0.2% succinate. Principal component analysis showed that the three independent replicates of each strain clustered closely (S3A Fig). Accordingly, differential expression analysis revealed that 1335 (727 upregulated and 608 downregulated) significant DEGs (FDR $P < 0.05$; 1.5-fold change cut-off) were identified in the Δ*pdhR* background relative to wild type EHEC (S1 Table and S3B Fig). The most strongly upregulated DEGs included D-alanyl-D-alanine dipeptidase *ddpX* (>205-fold; $P = 4.889 \times 10^{-224}$), carbon starvation protein *csiD* (>100-fold; $P = 1.151 \times 10^{-92}$), isocitrate lyase *aceA* (>74-fold; $P = 5.133 \times 10^{-241}$) and several ABC-type transporters (*ddpA*, *yhdX*, *argT* and *yhdY*). Conversely, the strongest downregulated genes included the majority of genes from the LEE pathogenicity island and the curli biosynthetic operon (*csgBAC;* >249-fold, >58-fold and >42-fold downregulated respectively) (Fig 3A). This data confirms that the regulatory effect of PdhR at the LEE1 promoter indeed results in downregulation of transcription across the entire LEE pathogenicity island, strengthening the case for PdhR as a major regulator of this locus. Furthermore, identification of differential expression of the curli operon suggests further roles for PdhR in regulating multiple virulence associated processes. Indeed, Gene Ontology analysis identified pathogenesis and protein secretion as the two most strongly enriched functional terms derived from the entire transcriptome, with other enriched terms corresponding to less specific processes such as translation and membrane composition likely mirroring the core function of PdhR (Fig 3B). Taken together, our data has identified PdhR as a major regulator of virulence processes in EHEC as highlighted by its defining role transcriptional activation of the LEE pathogenicity island and associated effector encoding genes.

## PdhR interacts directly with the LEE master regulatory region to activate T3SS expression

We next wanted to determine the mechanistic basis of PdhR regulation of the LEE-encoded T3SS. MEME analysis of the LEE1 master regulatory region revealed a sequence (tgTTGGTccttCCtgaT; conserved residues in uppercase; $P = 0.000673$) strongly resembling the AATTGGTnnnACCAATT consensus motif for PdhR derived from studies in *E. coli* K-12 (Fig 4A) [38]. To test this interaction, we overexpressed and purified 6xHis tagged PdhR (PdhR-his) for electrophoretic mobility shift assay (EMSA) analysis (S4A Fig), confirming that recombinant PdhR-his purifies as both a monomer and dimer in solution (S4B Fig) and is capable of binding to a known interaction site within its own promoter (S4C Fig). EMSA analysis using a 120 bp probe corresponding to DNA upstream of the LEE1 start codon revealed that PdhR-his was capable of directly binding to this region in vitro (Fig 4B). To confirm the specificity of this interaction, we generated a modified EMSA probe containing mutations in the critical residues of the PdhR motif (tgTTGGTccttCCtgaT -> tgTTGGTcctcttcagc). Mutation of this motif did not eliminate binding entirely but weakened the interaction between

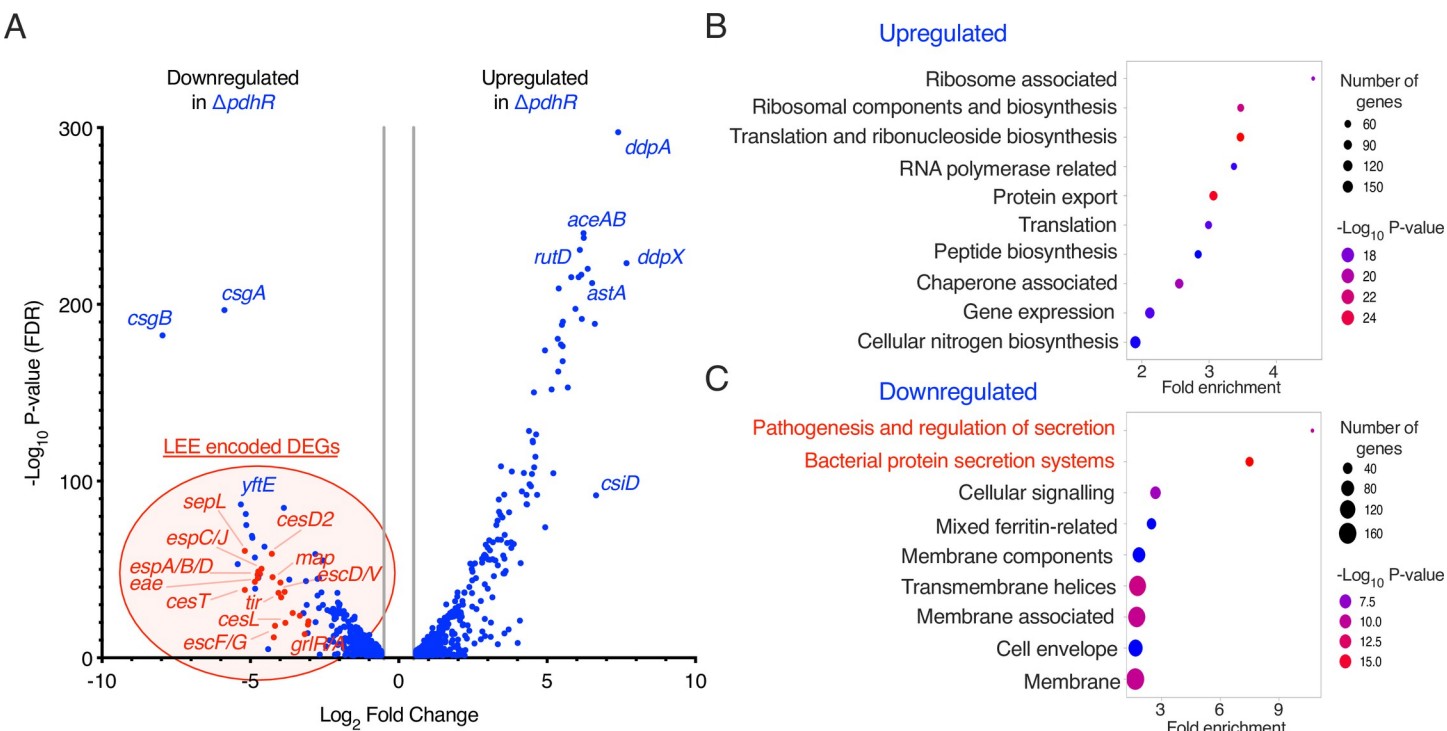

**Fig 3. PdhR is a global regulator of transcription in EHEC.** (A) Volcano plot depicting global gene expression patterns of Δ*pdhR* versus EHEC cultured in MEM-HEPES supplemented with 0.2% succinate, determined by RNA-seq. The data points represent all significantly differentially expressed genes (FDR *P* < 0.05) that were below the absolute fold-change threshold of +/- 1.5 (grey bars). The points highlighted in red are all LEE-associated. (B) Gene ontology analysis illustrating the most significantly enriched biological processes amongst the upregulated DEGs identified in panel A. (C) Gene ontology analysis performed on the downregulated DEGs from panel A, with LEE and virulence associated pathways highlighted in red.

PdhR and the LEE regulatory region in vitro, with the defined band shift no longer being observed using with 2 μM PdhR-his (Fig 4B). MEME analysis also identified a potential PdhR motif (tATgGaTagaACaAATT; conserved residues in uppercase; *P* = 0.000849) within the *grlA* promoter region, a gene that encodes a second master regulator of the LEE [12], however EMSA analysis of this region did not identify an interaction with PdhR-his (Fig 4B). Additionally, a negative control using a DNA probe corresponding to a fragment of the *amp* gene sequence gave us further confidence that the interaction identified between PdhR-his and the LEE1 promoter region was highly specific (Fig 4B).

The LEE1 regulatory region contains two promoters - the distal P1 and the proximal P2 [39–41]. P1 is known to be the major activation site of LEE1 transcription but seeing as the PdhR binding site was located in closer proximity to P2, we wanted to test at which promoter this TF was active. We engineered modified LEE1:GFP transcriptional reporter plasmids that correspond to either promoter individually and tested these in both wild type EHEC and the Δ*pdhR* backgrounds (Fig 4C). The analysis confirmed P1 as the major promoter controlling LEE transcription. Importantly, PdhR activity at P1 was almost identical to that using the LEE1:GFP reporter that corresponds to the full regulatory region (Fig 4D). Furthermore, there was no change in activity from the P2 promoter in the Δ*pdhR* background, suggesting that PdhR exerts its effect on the LEE via direct interaction with a binding site downstream of the P1 promoter. To absolutely confirm the functionality of this interaction in vivo, we generated a variant of the PLEE1:GFP reporter plasmid where we modified the PdhR binding site to match the mutated sequence used in our EMSA analysis (PLEE1$_{mut}$) where we observed a

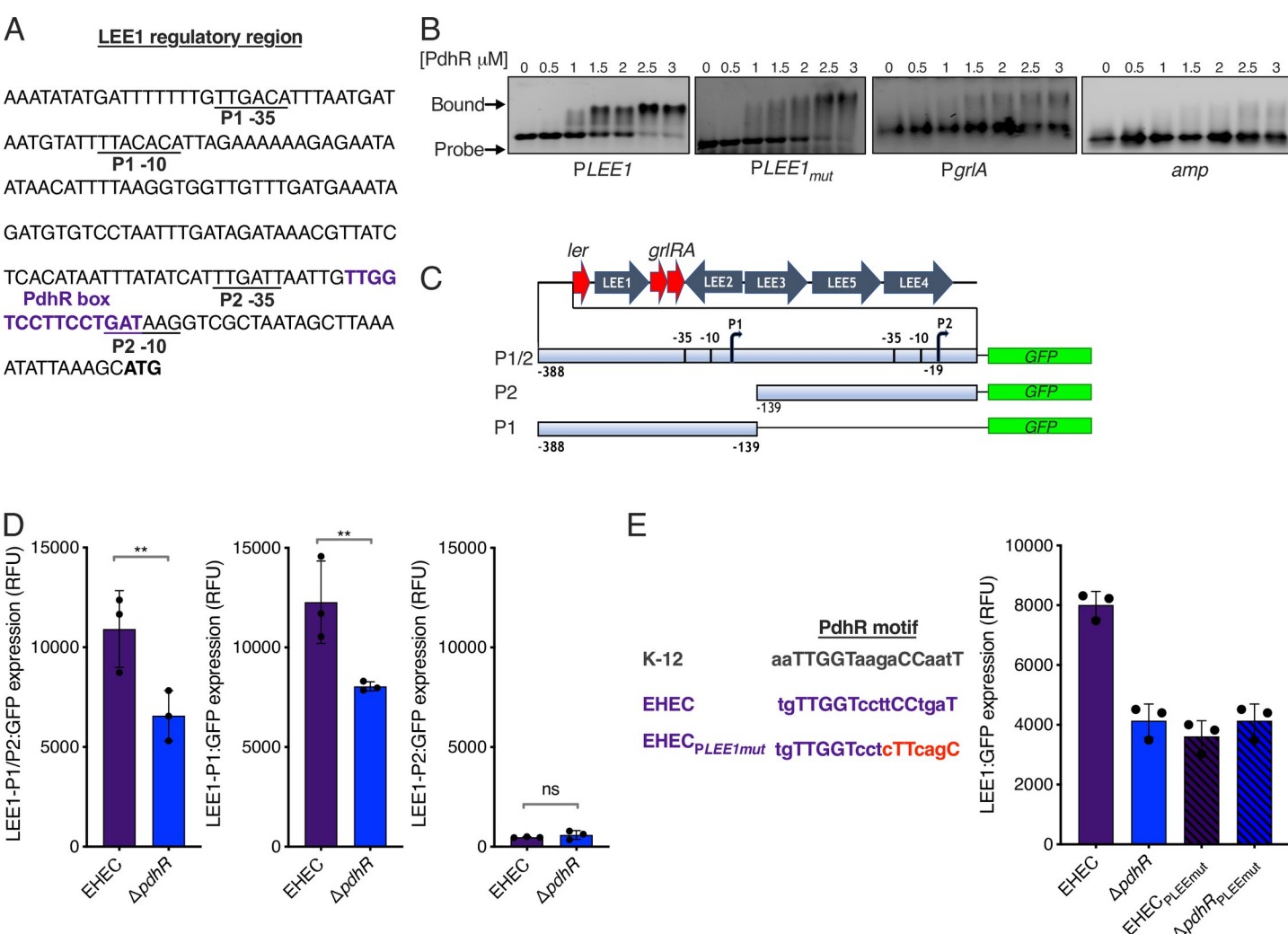

**Fig 4. PdhR binds to and directly regulates the LEE master regulatory region.** (A) Sequence of the LEE master regulatory region, directly upstream of operon LEE1. The *ler* start codon is highlighted in bold. Promoters 1 (P1) and 2 (P2) are underlined. The identified PdhR binding site is indicated in purple. (B) EMSA analysis of recombinant PdhR-his interactions with DNA probes corresponding to the *LEE1*, *LEE1mut* and *grlA* promoter regions. A negative control of the *amp* gene is shown also. Protein concentrations are indicated above each lane of the EMSA gel. EMSA experiments were performed on at least three independent occasions with reproducible results observed. (C) Schematic illustrating the design of various transcriptional reporter fusions of LEE promoter fragments to GFP, used to assess the activity of individual promoters located within the LEE regulatory region. (D) Transcriptional reporter assay of EHEC and Δ*pdhR* cultured in MEM-HEPES supplemented with 0.2% succinate. The graphs correspond to constructs designed in panel C (labelled P1/P2, P1 alone or P2 alone respectively). ** or ns indicate $P < 0.01$ or not significant, as determined by two-way ANOVA with Dunnett's post-test. Error bars represent standard error of the mean. (E) LEE1-GFP reporter assay of EHEC, Δ*pdhR*, EHEC_PLEEmut and Δ*pdhR*_PLEEmut. The sequence modification that _PLEEmut corresponds to, as well as the PdhR binding motif from *E. coli* K-12, are illustrated on the left of the graph. Error bars represent standard error of the mean.

weakened interaction (Fig 4B). This modification completely abolished PdhR-dependent activity at the LEE1 regulatory region, rendering promoter activity identical to that of the Δ*pdhR* mutant (Fig 4E). Collectively, these data demonstrate that PdhR exerts a positive effect on T3SS expression by directly interacting with a specific binding site located within the LEE1 master regulatory region.

## The PdhR binding site within the LEE regulatory region is widely conserved in A/E pathogens

We next aimed to assess how widely conserved this regulatory mechanism is across a wider range of *E. coli* isolates. To do this, we performed an analysis of the LEE1 master regulatory region (containing the identified PdhR binding site) from a selection of genomes consisting of the predominant EHEC serotypes (O157, O145, O111, and O26). The analysis aligned a 400 bp sequence upstream of the *ler* start codon from each strain. For the 168 strains analysed, the LEE1 promoter was found to be highly conserved with a mean normalised conservation score of 0.96 ±0.026 (S5A Fig). Within the O157 (n = 154) and O145 (n = 12) strains analysed, the PdhR box was completely conserved with no sequence variation (S5B Fig). The two O26 strains and single O111 isolate displayed more variation in their LEE1 region, with a sequence more closely related to that of *C. rodentium* but still contained an identifiable PdhR box. Therefore, the PdhR box was largely conserved in all EHEC serotypes and the murine A/E pathogen *C. rodentium* (S6C Fig). The high degree of conservation of the LEE1 region, and the PdhR binding region within it, suggests that the regulatory mechanism and associated phenotype that we identified in our prototypic TUV93-0 strain is expected to be shared with the predominant LEE-containing outbreak strains of EHEC.

Given that we identified a highly similar PdhR box in the *C. rodentium* LEE1 master regulatory region, we next wanted to test the effects of PdhR in this species given its utility as an in vivo model system for EHEC. We generated a *pdhR* mutant in the prototypical *C. rodentium* strain ICC168. Intriguingly, this deletion did not result in any observable fitness defects, highlighting potentially different metabolic circuitry between *C. rodentium* and *E. coli* (S6 Fig). Using RT-qPCR, we measured transcription of the LEE genes *ler* (LEE1), *espZ* (LEE2) and *eae* (LEE5) in the Δ*pdhR* background relative to wild type *C. rodentium*. This experiment showed that all three LEE-encoded genes were significantly downregulated when *pdhR* was deleted (Fig 5A). Similar to EHEC, this downregulation of the T3SS resulted in a significantly ($P < 0.0001$) attenuated ability to intimately attach to host cells and form A/E lesions, as was quantified using HeLa cell infection followed by fluorescence microscopy (Fig 5B and 5C). These results indicate that PdhR likely plays a role in controlling virulence more widely amongst different A/E pathogens, each with a distinct host range.

## PdhR contributes to *C. rodentium* colonisation of mice and regulates T3SS expression in vivo

The discovery that PdhR controlled the LEE-encoded T3SS in *C. rodentium* allowed us to address whether this phenotype affects pathogenesis during natural colonisation of a host. To test this, we orally infected female BALB/c mice ($N = 9$) with either wild type *C. rodentium* or Δ*pdhR*. We used colony counting from faeces as a measure of the infectious burden over a 20-day period. As early as 24 hours post-infection, the Δ*pdhR* mutant was significantly ($P < 0.01$) attenuated for faecal counts, indicating a colonisation defect relative to the wild type (Fig 6A). This colonisation defect was maintained throughout the peak of infection (day 8) and displayed the most significant difference of $> 2$ orders of magnitude ($P < 0.0001$) by day 17, with most Δ*pdhR* infected mice being cleared of the infection by day 20. The lack of a Δ*pdhR* growth defect in *C. rodentium* suggested that this colonisation defect occurred because of the decreased T3SS expression we observed in vitro but cannot rule out that, in the in vivo environment, loss of *pdhR* may be more reflective of a fitness defect. To test the former hypothesis, we purified RNA directly from colonic tissue of mice infected with either *C. rodentium* or Δ*pdhR* and used RT-qPCR to measure transcription of the *ler* gene in vivo. Despite normalisation, the levels of *ler* transcription were significantly ($\sim$4-fold; $P < 0.01$) lower in

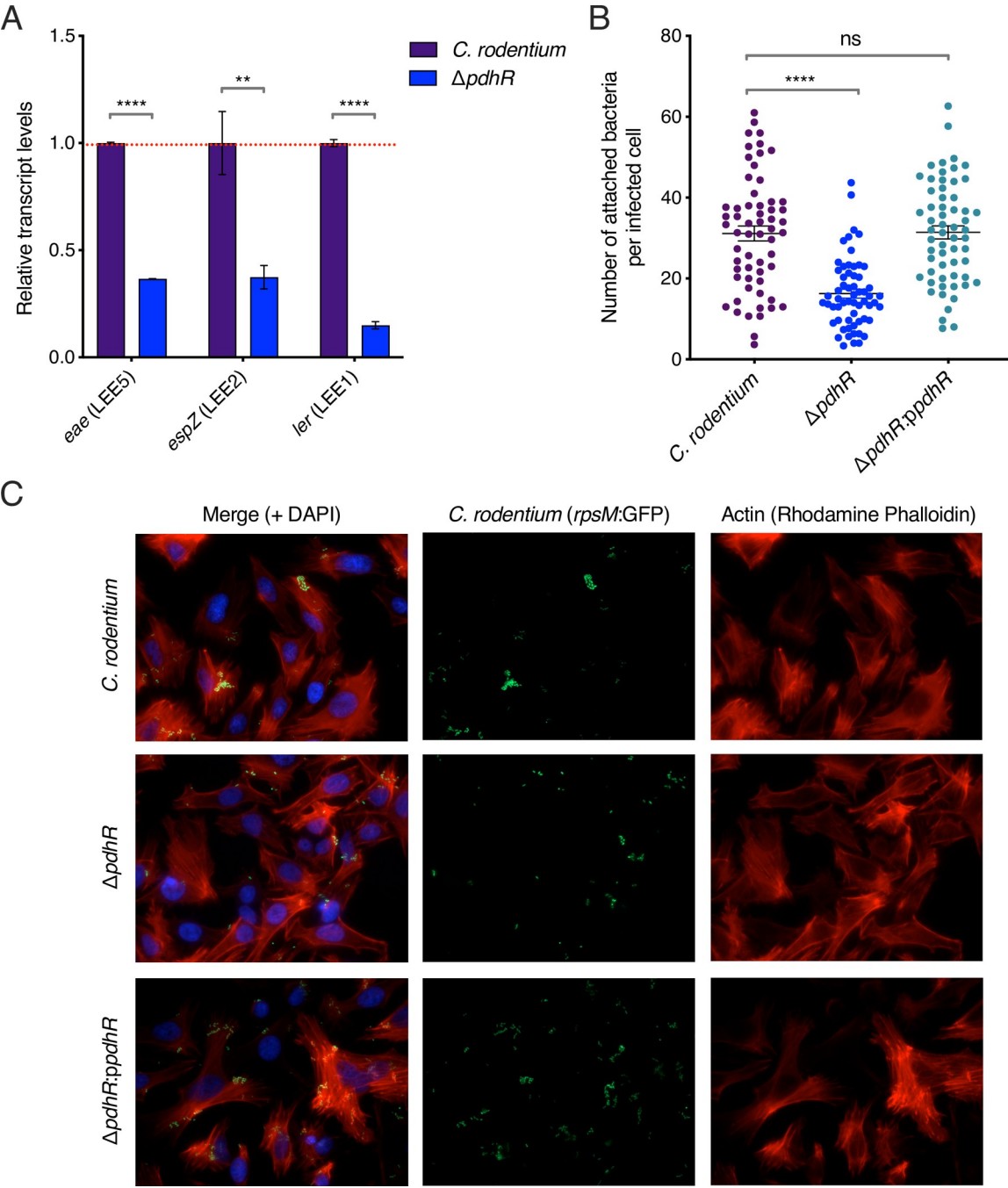

**Fig 5. PdhR promotes T3SS expression and host cell attachment in *C. rodentium*.** (A) RT-qPCR analysis of *eae*, *espZ* and *ler* from *C. rodentium* and Δ*pdhR* cultured in MEM-HEPES. Data were normalised to the *gapA* gene. \*\*\*\* or ns indicate $P < 0.0001$ or not significant respectively, as determined by two-way ANOVA with Dunnett's post-test. Error bars represent standard error of the mean. (B) Quantification of widefield fluorescence microscopy analysis of HeLa cells infected with *C. rodentium*, Δ*pdhR* and Δ*pdhR*:p*pdhR*. \*\*\*\* or ns indicate $P < 0.0001$ or not significant respectively, as determined by two-way ANOVA with Dunnett's post-test. Error bars represent standard error of the mean. (C) Representative images of HeLa cells infected with EHEC, Δ*pdhR* and Δ*pdhR*:p*pdhR*. The overlaid channels are labelled across the top row. The images represent a single field of view from $N = 10$ for each strain. This experiment was performed on three independent occasions.

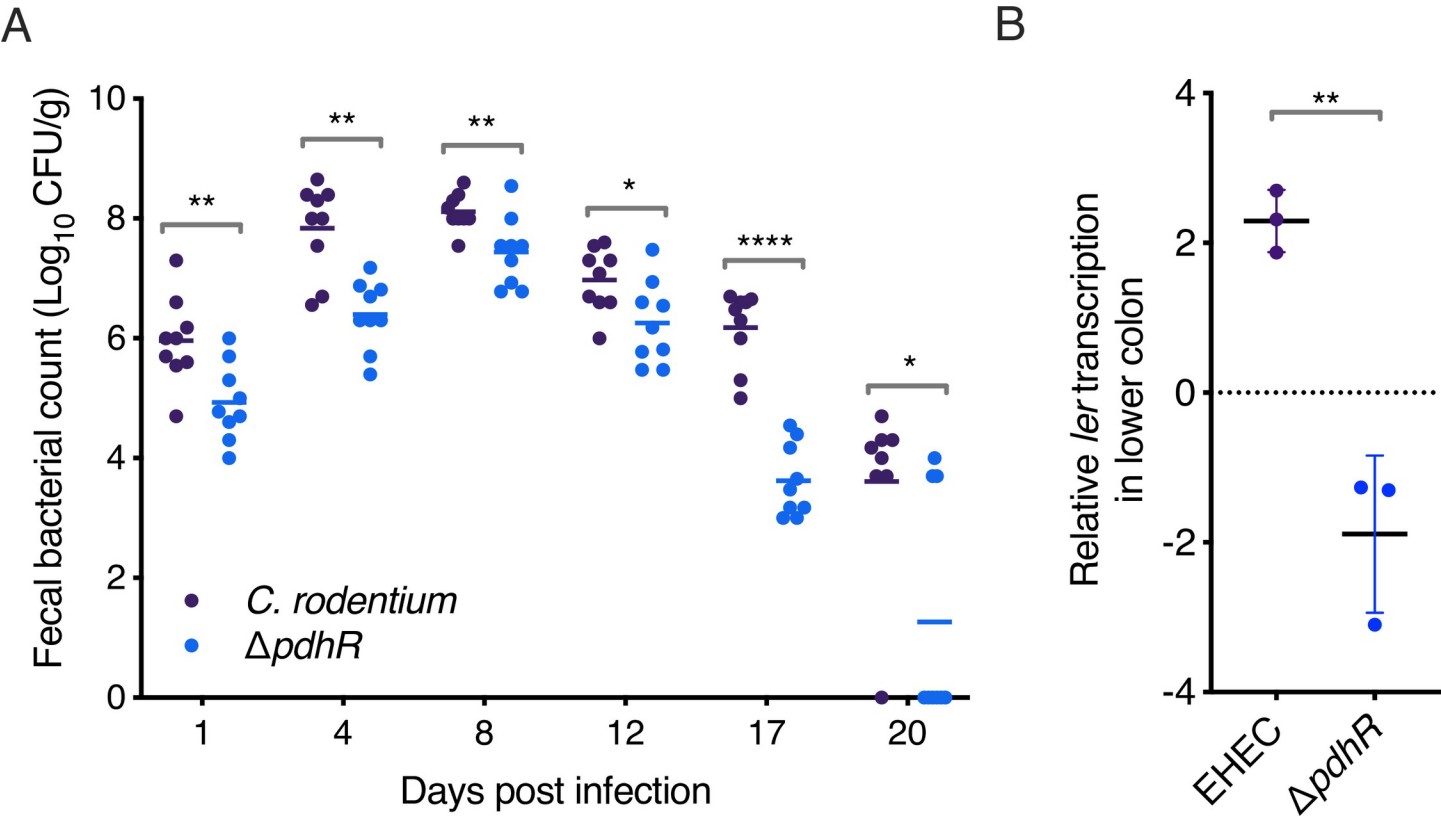

**Fig 6. PdhR contributes to the in vivo fitness of *C. rodentium* via T3SS regulation.** (A) Colonisation dynamics of female BALB/c mice (*N* = 9) orally infected with either *C. rodentium* or Δ*pdhR*. Colony forming units were determined from faecal samples taken at the indicated intervals. *, ** or *** indicate *P* < 0.05, *P* < 0.01 *P* < 0.0001 respectively, as determined by two-way Mann-Whitney *U*-test. Error bars represent standard error of the mean. (B) RT-qPCR analysis of relative *ler* transcription from *C. rodentium* or Δ*pdhR* infected colonic tissue. ** indicates *P* < 0.01, as determined by two-way ANOVA with Dunnett's post-test. Error bars represent standard error of the mean.

the Δ*pdhR* infected mice relative to the wild type, despite any effects that loss of this regulator may have on metabolic fitness (Fig 6B). Collectively, these data suggest that PdhR regulation of the LEE-encoded T3SS enhances the ability of A/E pathogens to colonise their hosts during infection.

## Discussion

The regulation of essential bacterial virulence factors ensures that they are deployed under spatial and temporal control in response to dynamically changing environments [3]. This allows pathogens to integrate the biochemical status of their surroundings with the regulation of genes that maximise their competitive fitness within the host. Furthermore, this network integration is often dependent on pre-existing TFs acquiring new roles in gene regulation [23]. Here, we have discovered that a central metabolic regulator, PdhR, directly controls virulence factor expression in the A/E pathogens EHEC and *C. rodentium*. PdhR binds to a precise site within the master regulatory region of the LEE-encoded T3SS, thus activating its expression. Deletion of *pdhR* results in a decrease in T3SS expression and a colonisation defect of both pathogens during interaction with host cells. Importantly, we found that this regulatory event is independent of the effects of PdhR on growth rate. Finally, we found that PdhR is required for maximal fitness within the murine gut and that deletion of *pdhR* coincides with a decrease of T3SS transcription in vivo.

One can speculate as to several potential sources of pyruvate within the host based on in vitro and in vivo evidence. During exponential growth in complex media, *E. coli* and other organisms export pyruvate derived from glycolysis or amino acid metabolism into the external environment in a process known as overflow metabolism [42–44]. This process tightly regulates the cytoplasmic pool of this integral metabolic intermediate and prevents toxicity. As availability of preferred carbon sources decrease and cells enter stationary phase, pyruvate (and other carbon sources) can be rapidly taken up from the external environment to facilitate further growth. Importantly, extracellular pyruvate is abundant in the host niche, the sensing of which may allow pathogens to respond to host inflammation or infection status [45]. It should be noted that the source of pyruvate within the host is not confined to microbial overflow metabolism as mammalian cells undergoing apoptosis (but not necrosis) have also been shown to actively produce pyruvate, the exploitation of which, drives expansion of *Salmonella* Typhimurium during murine infection [46].

The molecular mechanisms of extracellular pyruvate sensing and uptake during carbon starvation have been well described. This is coordinated by a dual two component sensor (TCS) network consisting of BtsSR and PyrSR. The higher affinity system BtsSR is activated by the cAMP-CRP complex resulting in expression of BtsT, a high affinity pyruvate/H+ symporter [47]. At concentrations greater than 600 μM, the PyrSR system activates YhjX, a predicted formate oxalate major facilitator superfamily antiporter thought to also transport pyruvate [48]. Importantly, the regulation of each of these TCS systems is co-ordinated with central carbon catabolism via cAMP-CRP-dependent activation of BtsSR, CsrA (carbon storage regulator A) dependent inhibition of BtsSR, and CsrA-dependent activation of YhjX. An additional transporter CstA is activated in stationary phase by cAMP-CRP to import pyruvate with moderate affinity [43]. Deletion of all three transporters is required to prevent growth on pyruvate as a sole carbon source. The redundancy of these systems highlights the importance of pyruvate sensing and homeostasis. The constant replenishment of the cellular pyruvate pool either through endogenous production (glycolysis, amino acid deamination) or uptake from exogenous sources would not only provide building blocks for cellular metabolism but also provide a signal of cellular carbon status that can be integrated into the control of virulence via PdhR. In addition, the promoter region of *pdhR* itself contains binding sites for BtsR, Cra and CRP (sourced from RegulonDB [49]), allowing fine tuning in response to sensing of exogenous pyruvate and fluctuations in central carbon metabolism.

Numerous studies have highlighted the pivotal role of pyruvate, not only as a central metabolic node connecting glycolysis to the TCA cycle, but also as a signal in governing virulence in diverse bacterial pathogens. For example, in *Yersinia pseudotuberculosis*, *Salmonella enterica* serovar Typhimurium and *Staphylococcus aureus*, pyruvate acts as a key control point, integrating metabolic adaptations with the regulation of virulence factors required for establishing infection [50–52]. Additionally, the T3SS of pathogenic *Plesiomonas shigelloides* has recently been shown to be activated by PdhR, although the precise mechanism remained to be determined [53]. In EHEC, endogenous and exogenous pyruvate enhance expression of the T3SS [32,33]. However, the mechanism by which sensing of pyruvate activates virulence in EHEC is not clear. Our data here suggests that pyruvate is sensed via PdhR to regulate the LEE. This is supported by the fact that exogenous pyruvate no longer enhances LEE expression in the absence of *pdhR*. Furthermore, loss of *pdhR* results in significant downregulation of the LEE under growth conditions that promote the accumulation of pyruvate. Importantly, the finding that it directly activates LEE expression during normal growth is distinct to its canonical function as a repressor of the pyruvate dehydrogenase complex operon. Indeed, recent genomic studies of the PdhR regulon in *E. coli* K-12 have identified its role as both an activator and repressor of global transcription, binding to more than 35 sites on the genome including

associated processes such as fatty acid degradation and motility [38,54]. Our transcriptome analysis in EHEC has further expanded our knowledge of PdhR's regulatory flexibility, by identifying a pathogen-specific regulatory mechanism that highlights the importance of PdhR in gene regulation beyond central metabolic processes.

Flexibility in core TF function has emerged as a major contributor to the regulation of bacterial virulence [23]. However, examples of more cryptic TFs that lack a discernible core metabolic role also exist. For example, we have extensively characterised the function of the LysR type YhaJ in distinct *E. coli* pathotypes. While YhaJ is highly conserved across the species, we found that it shares very little overlap in chromosomal binding sites and regulates completely different gene sets (including the LEE) between EHEC and uropathogenic *E. coli*. Furthermore, despite these important functions that seem to be tailored specifically to the infectious lifestyle, we still have not been successful in elucidating what the canonical role of YhaJ as a core-encoded TF is. Moreover, the reason for differences in binding sites and regulatory activities between strains is not clearly defined [29,55,56]. In the case of PdhR, it could be argued that given its importance as a regulator of central metabolism, this core TF is an ideal candidate for integrating virulence factor regulation with the normal function of the cell, irrespective of pyruvate levels. One hypothesis could be that PdhR maintains expression of the T3SS in line with the central nutritional requirements of the cell, whereas the activity of other TFs on LEE regulation play more of an accessory role to fine tune the T3SS in response to more specific environmental stimuli that are in a dynamic state of flux within the host niche.

Deletion of *pdhR* resulted in a fitness defect for EHEC, a phenotype that was successfully overcome by supplementation with succinate, similar to previous studies in *E. coli* K-12 [38]. This allowed us to separate fitness defects from the regulatory effect on the T3SS. Indeed, our transcriptomic analysis of Δ*pdhR* in EHEC revealed that several metabolic processes were dysregulated such as glycolysis, gluconate metabolism and the TCA cycle. However, *C. rodentium* Δ*pdhR* did not exhibit any growth defect in laboratory media. This left it unclear as to whether the in vivo fitness defect of Δ*pdhR* is due to downregulation of the LEE or interruption of a central metabolic regulatory circuit that may manifest as a fitness defect only during growth within the host gut. However, it is known that glycolytic flux is increased in the host during *C. rodentium* infection, thus providing exogenous pyruvate at higher concentrations, and that intimate colonisation results in oxygenation of the epithelial surface in a T3SS-dependent manner [57–59]. PdhR has also been shown to bind the promoters of several genes involved in aerobic respiration, including *ndh* and *cyoABCDE* encoding dehydrogenase II, and cytochrome *bo*-type oxidase respectively, further strengthening the ties between aerobic niche sensing and virulence regulation [34]. Together with our data showing that PdhR is a direct activator of the T3SS, and that LEE transcription is lower in the Δ*pdhR* mutant in vivo, this suggests that the observed defect in host colonisation throughout the infection cycle is at least partly driven by reduced T3SS activity.

In summary, we have discovered a new direct link between central metabolism and pathogenesis in A/E pathogens that highlights their essential requirement to maintain virulence factor regulation in parallel with the nutritional demands and energetics of the cell. Given our appraisal of the connections between pyruvate sensing and virulence regulation in several enteric pathogens, it is likely that this represents a generalised strategy for optimising fitness within the intestinal pathogenic niche.

## Materials and methods

### Ethics statement

All animal procedures were performed in strict accordance with the Animals in Scientific Procedures Act of 1986 with specific approval granted by the UK Home Office under PPL PI440270. The work was also considered and approved by the University of Glasgow Animal Welfare Ethical Review Body (AWERB) prior to implementation.

### Bacterial growth conditions

Bacterial strains were routinely cultured in lysogeny broth (LB) at 37 ˚C, 200 rpm. To select for plasmids, chloramphenicol, ampicillin or kanamycin was included at 25, 100 and 50 μg ml$^{-1}$ respectively. Assays requiring T3SS induction were conducted by diluting LB overnight cultures one-hundred-fold into prewarmed MEM-HEPES (Sigma; Cat #M7278) with appropriate antibiotics. Growth analysis was performed at 37 ˚C under aerobic conditions with shaking at 200 rpm. Specific growth rates ($\mu = \Delta$lnOD600 nm/$\Delta$t) were calculated as described previously [56]. A full list of strains and plasmids is provided in S2 and S3 Tables.

### Lambda red recombineering

In-frame deletions of TF-encoding genes *bssS*, *yfeC*, *rcnR* and *pdhR* were constructed using lambda red recombineering as outlined previously [60]. Briefly, the chloramphenicol resistance cassette from pKD3 was amplified with primers containing 5' 50 bp homology regions (red-F and red-R primers; S4 Table) for the target TF before column purification. TUV93-0 (pKD46) was cultured at 30˚C, 200 rpm in super optimal broth containing 100 μg ml$^{-1}$ ampicillin and 10 mM L-arabinose to an OD$_{600\ nm}$ of 0.4. The cells were harvested by centrifugation, washed three times in ice cold water and electroporated with one microgram of purified lambda red deletion cassette. Super optimal complete broth was added to the cell suspension and the cells were recovered at 30˚C, 200 rpm for two hours. The recovery mixture was plated on LB containing 25 μg ml$^{-1}$ chloramphenicol. Colonies were screened for successful integration of the chloramphenicol resistance cassette by PCR with primers whose annealing sites lay outside of the target region (for example 184-F and 184-R primers; S4 Table). To eliminate the resistance cassette and allow for transformation with chloramphenicol marked plasmids, each deletion cassette integrant was transformed with pCP20 and selected on LB with 100 μg ml$^{-1}$ ampicillin.

### Construction of plasmids

Plasmids were constructed using either restriction digestion followed by ligation or the NEBuilder method (New England Biolabs). For restriction enzyme digestion, 1 μg of purified DNA sample was incubated with 1 μl of each Fast-digest enzyme (New England Biolabs) and 2 μl of each buffer, brought to a total volume of 20 μl with nuclease free water. These were incubated for 1 h at 37˚C unless otherwise stated. Digestion reactions was separated on a 0.8% agarose-TAE gel, and samples were purified using the QIAquick gel extraction kit (QIAGEN) as per manufacturer's instructions. Digested DNA was ligated into linear plasmids at a 2:1 insert to vector ratio. 1 μl of T4 ligase (New England Biolabs) and 2 μl of T4 buffer were added and made up to a reaction volume of 10 μl with nuclease free water. These reactions were incubated for 2 hours at room temperature. 5 μl of sample was used for transformation into 50 μl of competent cells by heat shock. After transformation recovery the culture was plated onto LB agar plates containing the relevant antibiotic for selective screening. Positive colonies were confirmed by PCR amplification of the cloned region and DNA sequencing (Eurofins). For

NEBuilder, the online assembly tool (New England Biolabs) was used to design primers for assembly-based cloning. Insert sequences were amplified by PCR with the relevant overhangs and the vector backbone linearised by PCR. Assemblies were carried out using a Gibson Assembly Cloning Kit (New England Biolabs) according to the manufacturer's instructions; 0.5 pmol of vector to insert at a ratio of 1:1, 10 μl of NEBuilder HiFi DNA Assembly Master Mix, and nuclease free water to a total volume of 20 μl. Samples were set up on ice then incubated in a thermocycler at 50°C for 15 minutes. 2 μl of the sample was transformed into competent cells. As above, positive colonies were confirmed by PCR amplification of the cloned regions and DNA sequencing (Eurofins).

### GFP-promoter fusion reporter assays

Transcriptional reporter analysis was performed using strains TUV93-0 (EHEC) and ICC169 (*C. rodentium*) transformed with transformed with the relevant pAJR71-based promoter fusion constructs [61]. Reporter strains were grown overnight in LB (with chloramphenicol 20 μg/ml) and 50 μl of this culture was then inoculated into 5 mL pre-warmed MEM-HEPES and grown to an optical density of ∼0.8 (late log phase) at 37°C with 200 rpm shaking. To measure GFP expression from reporters, 200 μl of the culture was transferred to a black flat bottom 96 well microtiter plate and analysed using a FLUOstar Optima Fluorescence plate reader (BMG, Labtech, UK). Background fluorescence was measured using the respective strains containing the promotorless version of the pAJR71 construct. Relative expression was then calculated as absolute GFP output divided by the optical density at equivalent timepoints. Experiments were performed in biological triplicate, each carried out on a separate occasion.

### SDS-PAGE profiling of secreted proteins

Culture supernatants of EHEC strains grown in MEM-HEPES were normalised by optical density and collected by centrifugation at 3,750 g for 10 minutes before separation form the cell pellet. The supernatant was filtered through a 0.2 μM filter before ice cold Trichloroacetic acid (Sigma) was added to a final concentration of 10% (v/v). 1 μL of lysozyme (2 mg/mL) was added to each sample, as a co-precipitant for maximum protein recovery and to act as a loading control. Samples were precipitated at 4°C overnight before the secreted proteins were collected by centrifuging the samples for 30 min at 6000 g. The supernatant was carefully removed, and the tubes were inverted and allowed to air dry for 15 minutes. The pellet was resuspended in 25 μL of 1x LDS sample buffer (Thermo Fisher), boiled at 97°C for 10 min and analysed by SDS-PAGE on a 4–12% Bis-Tris protein gel using the NuPage system (Invitrogen). SeeBlue Plus2 protein standard (Invitrogen) was ran alongside the protein samples. The samples were separated by electrophoresis at 160 v for 30 minutes. Gels were stained with Coomassie blue for 1 hour and de-stained overnight in double distilled water. De-stained SDS-PAGE gels were then visualized using a ChemiDoc MP Imaging System (Bio-Rad).

### HeLa cell adhesion assays and fluorescence microscopy

Cultures of *rpsM*:GFP containing EHEC and *C. rodentium* with their respective Δ*pdhR* and Δ*pdhR*:p*pdhR* derivative strains were grown in MEM-HEPES supplemented with 0.2% succinate to an optical density of 0.4 before back diluting to 0.1 with MEM-HEPES. HeLa cells were seeded (104 cells per well in 24-well plates) on sterile coverslips in MEM-HEPES with 10% foetal calf serum. One hour prior to infection cells were washed three times with PBS and fresh media was added with any supplementary additions. A volume of 100 μl of the above bacterial culture was added per well. The plates were centrifuged at 300 g for three minutes and incubated at 37°C for 3 hours. After incubation, the cells were washed five times with PBS, fixed

with 4% paraformaldehyde and incubated for 15 min. Next, cells were permeabilized with 0.1% triton X-100 for 10 minutes before washing and staining for 1 hour with Rhodamine phalloidin-Alexafluor (Invitrogen). Coverslips were washed before mounting and analysing using a Zeiss Axioimager M1 and Zen Pro software. A/E lesions were identified by condensation of host actin around the site of bacterial attachment. Host cell-associated bacteria were quantified using the event counter tool in Zen and the total percentage of infected cells was also determined from at least 10 random fields of view across 3 cover slips.

## Animal colonisation experiments

Specific pathogen free female BALB/c mice age 6 weeks were used for all experiments. Wild type *C. rodentium* ICC169 and Δ*pdhR* were grown in MEM-HEPES to an optical density of ∼0.7 before being centrifuged and resuspended at 100x concentration in PBS. Mice were then inoculated by oral gavage with 200 μl of the bacterial suspension (∼3 x $10^9$ CFU). Mice were checked daily for weight loss and signs of any adverse effects. The infection experiments were performed in two independent cohorts of mice. For analysis of CFU counts, stool samples were recovered aseptically and homogenized in PBS before serial dilution. The number of viable CFU per gram of stool was determined by plating onto LB agar containing the appropriate antibiotic. Tissues for RNA extraction were collected and immediately immersed in RNAlater stabilisation reagent then placed on ice (Ambion).

## RNA extraction for bacterial culture and murine tissue

RNA was extracted from triplicate bacterial cultures using a PureLink RNA Mini Kit (Thermo Fisher Scientific) according to the manufacturer's instructions. Cell pellets were resuspended in 100 μl of TE buffer containing lysozyme (10 mg/ml) with 0.5 μl of 10% SDS solution before vortexing. The samples were incubated for 5 min at room temperature, then 350 μL of lysis buffer containing 1% B-mercaptoethanol was added to each tube and the samples were mixed again by vortexing. 250 μL of 100% ethanol was added to the samples and then they were vortexed to remove any visible precipitate. Samples were transferred to spin columns, washed according to the PureLink protocol and RNA was eluted in nuclease free water. For in vivo samples, infected tissue RNA was extracted using an RNeasy Mini Kit as per the manufacturer's instructions. Briefly, snap frozen tissues were thawed on ice, residual fluid and blood from the tissue was blotted off. 750uL Qiazol and one 7 mm stainless steel ball was added to each samples and homogenisation was performed using a Tissue Lyser LT (Qiagen) at 25 Hz (250 frequency) for 2 minutes. 150uL chloroform was added to samples and vortexed for 10 seconds, following incubation at room temperature for 5 minutes. Samples were centrifuged for 15–30 minutes at 12000 x g. The upper clear phase was collected and transferred to a fresh tube and 1.5 volumes of 100% ethanol was added. The mixture was briefly vortexed for 15 seconds followed by purification through an RNeasy column and elution in nuclease free water. Turbo DNase treatment (Thermo Fisher) was used to remove residual DNA from the sample, as per manufacturers instructions. Final RNA samples were purified by phenol-chloroform extraction and ethanol precipitation and concentration was measured using a NanoDrop spectrophotometer.

## Quantitative real time PCR (RT-qPCR)

Ten nanograms of total RNA was used as a template to prepare 10 μL cDNA using LunaScriptRT SuperMix (New England Biolabs) according to manufacturer's instructions. cDNA samples were diluted 1 in 20 prior to RT-qPCR using a Luna Universal qPCR Master Mix (New England Biolabs). RT-qPCR was performed on a CFX-Connect real-time PCR detection

system (Bio-Rad). Prior to RT-qPCR, the primer efficiency of each primer pair was evaluated using the equivalent thermocycling conditions and a series of concentration standards prepared from template gDNA (100, 4, 0.8, and 0.16 ng/μL). Only primers with efficiencies between 90 and 110% were selected. Analysis was performed using CFXConnect software (Bio-Rad), using the *gapA* housekeeping gene a calibrator. Relative expression was determined using the 2-ΔΔct method [62].

## RNA-sequencing and transcriptome analysis

RNA sample quality was assessed using an Agilent Bioanalyzer 2100. Ribosomal depletion was carried out using a MICROBExpress kit (Thermo Fisher) according to the manufacturer's instructions. Library preparation and sequencing was carried out at the University of Glasgow Polyomics facility. Sequencing libraries were prepared with a TrueSeq Stranded mRNA Library Prep kit (Illumina). Sequencing was carried out on the Illumina NextSeq 500 platform with at least 10 million 100 bp single end reads being obtained. FastQC (Babraham Bioinformatics) was used to assess the quality of raw reads (minimum Phred threshold of 20). Raw data was mapped to the EDL933 genome and plasmid reference (NCBI accession number: NC_002655) using CLC Genomics Workbench 20. Differential expression analysis was done using EdgeR, implemented in CLC Genomics Workbench. Corrected *P*-value threshold (false discovery rate) was set at 0.05 and absolute fold change of greater than +/- 1.5 was considered significant. Normalized RNA-seq data transcript quality assessment was determined using principal component analysis (PCA) plots to visualize relationships between samples. The RNA-seq raw data has been deposited in European Nucleotide Archive (https://www.ebi.ac.uk/ena/browser/home) under the accession number PRJEB74273.

## Protein overexpression and purification

The *pdhR* coding sequence was cloned into pET21a(+) as described above and transformed into BL21 DE3 cells. For overexpression, 2 litres of LB was inoculated with this and cultured until an optical density of 0.5 was reached. At that stage 0.5 mM IPTG was added, and the cultures were grown overnight at 30˚C. The cell pellet was harvested by centrifugation and resuspended in protein buffer A (50 mM Tris, 0.5 M NaCl, and 5% glycerol) containing lysozyme, EDTA-free protease inhibitor mix, and DNase (Promega). Cells were lysed by sonication on ice, with 1 second on and 1 second off rotations for 6 minutes followed by centrifugation for 50 minutes at 4˚C, 18000 rpm. The supernatant was removed and filtered through a 0.22 μM filter before being applied to a Ni2+ chelating column (HisTrap High Performance, GE Healthcare) equilibrated with protein buffer A containing 5 mM imidazole. Recombinant PdhR-his was eluted with a linear gradient from 10 to 300 mM imidazole added to buffer A. Fractions containing PdhR-his were pooled, dialyzed against buffer A and stored at -90˚C in buffer A.

## Immunoblot analysis

Proteins were transferred from SDS-PAGE gels to a 0.45 μM nitrocellulose membrane (Thermo Fisher) using the NuPage transfer system (Invitrogen) at 30 volts for 1 hour. 5% skim milk powder in PBST was used to block the membrane. Blocked membrane was incubated with primary antibody (Anti-6x His; Sigma) at a 1/5000 dilution in 1% skim milk/PBST solution for 1 hour at room temperature with gentle agitation. The membrane was washed three times with 50 ml PBST then incubated for 1 hour with secondary antibody (HRP-conjugated; Sigma) before being washed in PBST. Finally, the membranes were developed using Pierce ECL Plus Substrate for 5 mins and imaged using a ChemiDoc MP imaging system (BioRad).

## Electrophoretic mobility shift assays (EMSA)

EMSA assays were performed using fragments of target promoter regions amplified by PCR using genomic DNA of EHEC TUV93-0 as a template. The resulting PCR fragments were purified, and 150 nM samples were incubated on ice in binding buffer (250 mM Tris-HCl pH 8, 0.5mM EDTA, 25 mM MgCl, 25 mM MgCl, 5 mM DTT, 25% Glycerol) with increasing amounts of purified PdhR-his and 0.5 µg/ml of poly dIdC in a total volume of 16 µl. After careful mixing, samples were incubated for 20 min at 37˚C, placed back on ice for 10 min, and then loaded onto 2% agarose gel in 0.5x TBE. Electrophoresis was carried out in 0.5x TBE at 50 volts, 4˚C for 4 hours. Finally, the gel was stained with gel red and imaged using a ChemiDoc MP (Bio-Rad).

## Site directed mutagenesis

Site directed mutagenesis for EMSA probes was carried out using the Q5 Site-Directed Mutagenesis Kit (New England Biolabs) according to the manufacturer's recommendations. Briefly, primers were designed and annealing temperature calculated using the NEBaseChanger online tool). Substitutions are designed by including the desired nucleotide change in the centre of the forward primer, including at least 10 complementary nucleotides on the 3´end of the mutation. The reverse primer is designed so that the 5´ ends of the two primers anneal back-to- back. While deletions are designed by engineering standard, non-mutagenic forward and reverse primers that flank the region to be deleted. Then the assembly mix was prepared as per the manufacturer's instructions using $\sim$25 ng of template DNA and 0.5 µM of each primer. The assembly mix run using a standard PCR machine. 1 µl of the amplified PCR product was then incubated for 5 mins with 1 µl 10x KLD enzyme mix (containing a kinase, a ligase and DpnI), 5 µl 2x KLD reaction buffer and nuclease free water, allowing for rapid ligation of the PCR product and removal of the template DNA. Then 5 µl of the mix was transformed into competent cells using the standard heat shock procedure. 100 µL of the recovered cells were spread on LB agar containing the appropriate antibiotic. Mutagenised inserts were verified by PCR amplification followed by sequencing (Eurofins).

## Sequence conservation analysis

To examine the conservation of the LEE1 master regulatory region, we downloaded a set of complete Refseq assemblies spanning the O157, O145, O111 and O26 serotypes–as well as including complete *C. rodentium* assemblies (S5 Table). We reannotated these based on the *E. coli* O157:H7 proteome (NCBI Accession number GCF_000008865.2) to ensure consistency across annotations, using Prokka (Galaxy EU Server, Version 1.14.6+galaxy1) to generate gff3 formatted files and corresponding fasta formatted files [63,64]. In R (v4.3.2; RStudio v2024.04.2), the gff3 files were used to extract 400 bp upstream from annotated *ler* genes, corresponding to the LEE1 promoter region [65]. This was achieved using the Biostrings (v2.70.3) and BSgenome (v1.70.2) packages [66,67]. The conservation of this region was scored using the msaConservationScore function as implemented in msa (v1.34.0), with the default nucleotide substitution matrix [68]. Normalised conservation scores were calculated by dividing each score by the maximum score of the promoter region. These were plotted using ggplot2 (v3.5.1) [69,70]. To examine the phylogenetic conservation of the promoter region, we used Roary (Galaxy EU Server, v3.13.0+galaxy2) to generate a core-gene alignment from the Prokka annotations. This alignment was used to build a maximum-likelihood consensus tree using IQ-TREE (Galaxy EU Server, v2.3.3+galaxy0) with 1234 bootstrap replicates [71]. This tree was visualised using the following packages: treedataverse (v0.0.1), matchmaker (v0.0.1), ggtree (v3.10.1), ggtreeExtra (v1.12.0), and ggnewscale (v0.4.10), and annotated with the LEE1

promoter region using the gheatmap function [72–76]. To identify the conservation of the PdhR Box, the Clustal algorithm as implemented using the msa package in R and was used to align 400 bp of a representative strain from each LEE-positive serotype analysed. The msaR (v0.6.0) alignment browser was used to identify the pdhR1 box in this alignment, and the sequence logo visualised with ggmsa (v1.6.0) [77–79]. Genome accessions numbers for download from NCBI are listed in S5 Table.

## Statistical analyses

Statistical analysis was performed using GraphPad Prism (Version 8) unless otherwise stated. RT-qPCR analysis was carried out using CFX-Connect software, according to the 2-ΔΔct method. RNAseq data processing and analysis was computed using CLC Genomics Workbench. For TF binding site identification, the consensus recognition sequence for PdhR, which consists of a 17 bp palindromic sequence comprised of AATTGGTnnnACCAATT, was used. Prediction of binding sites from EHEC sequences was carried out using the MEME-Suite (version 5.1.0) under default parameters.

## Supporting information

**S1 Fig. PdhR is required for pyruvate-associated LEE regulation.** LEE-GFP reporter assay of EHEC and Δ*pdhR* cultured in MEM-HEPES alone or MEM-HEPES supplemented with 1 mM pyruvate. * and ns indicate $P < 0.05$ or not significant respectively, as determined by two-way ANOVA with Dunnett's post-test. Error bars represent standard error of the mean.
(TIF)

**S2 Fig. Deletion of *pdhR* results in a growth defect that can be alleviated by supplementation with succinate.** Specific growth rates were calculated between two and five hours using data from Fig 2A. Bars indicate means of three experiments with each experimental observation indicated by data points. Statistical significance was assessed by Ordinary one-way ANOVA with Tukey's post-test for multiple comparisons (* and ns indicate $P < 0.05$ or not significant respectively).
(TIFF)

**S3 Fig. Transcriptome analysis of Δ*pdhR* versus wild type EHEC.** (A) Principal component analysis of the Δ*pdhR* versus EHEC transcriptomic data determined by RNA-seq., verifying the clustering of independently prepared biological replicates. (B) Total number of up and downregulated DEGs identified by RNA-seq.
(TIFF)

**S4 Fig. Purification of recombinant 6xhis-tagged PdhR.** (A) SDS-PAGE analysis of over-expressed recombinant PdhR-his. The gel reveals elution of monomeric or dimeric PdhR in solution. (B) Immunoblot analysis of selected fractions from panel A confirming the bands as corresponding to monomeric or dimeric PdhR. (C) EMSA analysis of purified PdhR-his in complex with its own promoter region (P*pdhR*), verifying the functionality of the recombinant protein.
(TIF)

**S5 Fig. The PdhR binding site within the LEE master regulatory region is highly conserved across EHEC and *C. rodentium* isolates in nature. (A)** Per-position conservation score of the unaligned upstream regions from all 168 assemblies in the dataset. (B) Core genome maximum-likelihood consensus tree of the 168 isolates queried here, assembled from an alignment of 707 core genes generated by Roary. This is lined-up with the unaligned LEE1 master

regulatory region (upstream region of the *ler* start codon), illustrating how conserved the promoter region is amongst EHEC isolates. The non-LEE carrying EC-10 is included as a negative control. (C) ClustalW alignment of the region upstream of *ler* in *C. rodentium* and selected isolates of the *E. coli* O157, O145, O111, and O26 serotypes. The PdhR box is highlighted in red and the sequence logo of this motif is illustrated above the aligned sequence.
(TIF)

**S6 Fig. Deletion of *pdhR* has no effect on growth in *C. rodentium*.** Growth curves depicting hourly optical density (600 nm) measurements of *C. rodentium* and the Δ*pdhR* mutant cultured in MEM-HEPES. Error bars represent standard error of the mean.
(TIF)

**S1 Table. List of up- and down-regulated genes identified in the Δ*pdhR* background relative to wild-type EHEC.**
(XLSX)

**S2 Table. Bacterial strains used in this study.**
(DOCX)

**S3 Table. Plasmids used in this study.**
(DOCX)

**S4 Table. Primers used in this study.**
(DOCX)

**S5 Table. NCBI accession numbers of LEE-encoding isolates analyzed in this study.**
(DOCX)

**S1 Source data. Raw data used to generate the figures in this study.**
(XLSX)

## Acknowledgments

We are grateful to Glasgow Polyomics for the RNA-seq library preparation and sequencing.

## Author Contributions

**Conceptualization:** James P. R. Connolly, Andrew J. Roe.

**Formal analysis:** Ester Serrano, David R. Mark, James P. R. Connolly.

**Funding acquisition:** Gillian R. Douce, Andrew J. Roe.

**Investigation:** Kabo R. Wale, Nicky O'Boyle, Rebecca E. McHugh, Ester Serrano.

**Supervision:** Nicky O'Boyle, Gillian R. Douce, James P. R. Connolly.

**Writing – original draft:** Nicky O'Boyle, James P. R. Connolly, Andrew J. Roe.

**Writing – review & editing:** Kabo R. Wale, Nicky O'Boyle, Rebecca E. McHugh, David R. Mark, Gillian R. Douce, James P. R. Connolly, Andrew J. Roe.

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
