## [Decision Letter · Decision Letter 0]

6 Sep 2024

Dear Dr Roe,

Thank you very much for submitting your manuscript "A master regulator of central carbon metabolism directly activates virulence gene expression in attaching and effacing pathogens" for consideration at PLOS Pathogens. As with all papers reviewed by the journal, your manuscript was reviewed by members of the editorial board and by several independent reviewers. The reviewers appreciated the attention to an important topic. Based on the reviews, we are likely to accept this manuscript for publication, providing that you modify the manuscript according to the review recommendations.

Sincerely,

John M Leong

Academic Editor

PLOS Pathogens

Thomas Guillard

Section Editor

PLOS Pathogens

Michael Malim

Editor-in-Chief

PLOS Pathogens

orcid.org/0000-0002-7699-2064

Reviewer Comments (if any, and for reference):

Reviewer's Responses to Questions

**Part I - Summary**

Reviewer #1: The manuscript describes how a central regulator of metabolism can be integrated into the virulence gene regulatory network of attaching and effacing pathogens. The regulator, PdhR, which controls gene expression in response to pyruvate availability within the intestine, directly regulates genes encoding a type III secretion system in enterohemorrhagic E. coli and the mouse pathogen Citrobacter rodentium.

Using a transcriptomics approach the authors find that PdhR controls expression of the global regulator Ler of the LEE PAI. They show in both EHEC and C. rodentium that LEE gene expression is diminished in a pdhR knockout strain, and that A/E lesions are reduced in a tissue culture model using C. rodentium. They show that purified PdhR protein binds to a conserved PdhR binding site and that colonization of the mouse intestine is less compared to wild type in the pdhR deletion strain.

In sum, the work gives a specific example as to how enteric pathogens can integrate central metabolism and host signaling events into virulence gene expression, which contributes to disease. The work is novel, important to the field, and draws clear conclusions based on multiple lines of evidence. A connection to central metabolism in the study of molecular pathogenesis is a critical part of our understanding, and this work pushes knowledge forward in this key area of inquiry.

Reviewer #2: This manuscript demonstrates that the TF PdhR is a regulator of the T3SS system in EHEC and C. rodentium. The authors previously reported that pdhR expression was increased in C. rodentium during mouse colonic infection. Here, the authors built on this finding to demonstrate a role for PdhR in regulating expression of ler, the master regulator of the T3SS-encoding LEE genes. Overall, this is a well-written manuscript that uses complementary approaches to justify conclusions. The findings are significant as the findings provide new insights into regulatory mechanisms important for EHEC and C. rodentium virulence. The findings also link metabolism and virulence. I have a couple of points that the authors should address:

1. A competition experiment for the EMSAs should be included. This is the gold-standard for demonstrating specificity of binding.

2. How were the data normalized for the in vivo C. rodentium experiments showing ler expression? I did not see this described in the methods. The authors report decreased recovery of the pdhR strain compared to wt, and this would certainly lead to lower ler transcripts levels. The RT-qPCR are shown as relative values. What were the ct values for gapA? Differences in gapA transcript levels would affect relative ler expression using the analysis method described in the text.

3. Please include the C. rodentium pdhR growth curve in the supplemental data.

Reviewer #3: This is a strong manuscript and very straightforward showing for the first time that the PdhR transcription factor that regulates pyruvate utilization in E. coli also regulates transcription of the LEE region and EHEC and C. rodentium, as well as virulence and pathogenesis during murine infection. This is important information to the field.

There are just some very minor weakness that are easily addressable with text changes or adding some extra information. I detailed them below.

**Part II – Major Issues: Key Experiments Required for Acceptance**

Reviewer #1: None.

Reviewer #2: 1. A competition experiment for the EMSAs should be included. This is the gold-standard for demonstrating specificity of binding.

2. How were the data normalized for the in vivo C. rodentium experiments showing ler expression? I did not see this described in the methods. The authors report decreased recovery of the pdhR strain compared to wt, and this would certainly lead to lower ler transcripts levels. The RT-qPCR are shown as relative values. What were the ct values for gapA? Differences in gapA transcript levels would affect relative ler expression using the analysis method described in the text.

3. Please include the C. rodentium pdhR growth curve in the supplemental data.

Reviewer #3: No major issues or key experiments needed

**Part III – Minor Issues: Editorial and Data Presentation Modifications**

Reviewer #1: 1. Line 149. The authors use 1 mM pyruvate supplemented MEM-HEPES to illustrate PdhR-mediated signaling. Is the concentration of pyruvate in the mouse, or human intestine known, and how might this affect signaling within the intestine?

2. Beginning on line 156, the authors discuss the previously observed growth defect caused by pdhR deletion in E. coli. While the colonization data presented in Figure 6 reach significance, how are the authors certain that the change in CFU’s per g of intestinal tissue don’t arise because of a growth defect of the deletion strain? While the authors also show diminished LEE gene expression, could a combination of events be occurring? Is a varied gene regulatory circuitry in C. rodentium compared to EHEC of concern with the observed results?

3. Line 207. It is unclear what the authors are referring to by the phrase “translation membrane composition,” “with other enriched terms corresponding to less specific processes such as translation membrane composition likely mirroring the core function of PdhR.” Please clarify.

Reviewer #2: n/a

Reviewer #3: The emphasis on pyruvate regulation of LEE expression through PdhR should be toned down a bit. The only data here is Figure S1 and although significant it is not that striking. The data on PdhR regulation per se is very strong.

The authors should add more details in methods on the environmental conditions the cultures were grown. I got 37oC in MEM-HEPES. However, it is important to add the growth phase (early, mid, late log or stationary?) as well as oxygen tension (aerobic, anaerobic or microaerobic conditions). All of these are known to affect LEE expression, and it is important for the community to know when this regulation is occurring.

Growth curves on Figure 2A need p Values and the calculation of doubling rates

Can the authors please clarify if the subsequent EHEC experiments were performed with or without succinate?

Fig. 2D, The first FAS panel is great, I don't see the need for the GFP or Rodamine panels, instead it would be batter to have an inlet or an extra panel of a higher magnification of the FAS images.

Fig. 5A would be nice to add the complement to the transcription studies, since they obviously have that on the FAS experiments

Fig. 5C on FAS same comment as per FAS on Fig. 2

On animal experiments please state clearly how many cohorts of animals were used

PLOS authors have the option to publish the peer review history of their article (what does this mean?). If published, this will include your full peer review and any attached files.

Reviewer #1: No

Reviewer #2: No

Reviewer #3: No

Figure Files:

Data Requirements:

Reproducibility:

References:

---

## [Editor Report · Decision Letter 1]

24 Sep 2024

Dear Andy,

Thank you very much for submitting your manuscript "A master regulator of central carbon metabolism directly activates virulence gene expression in attaching and effacing pathogens" for consideration at PLOS Pathogens. As with all papers reviewed by the journal, your manuscript was reviewed by members of the editorial board and by several independent reviewers. The reviewers appreciated the attention to an important topic. Based on the reviews, we are likely to accept this manuscript for publication, providing that you modify the manuscript according to the review recommendations: For clarity, please move the experiment described in Fig. S5 into Fig. 4.  

Sincerely,

John

John M Leong

Academic Editor

PLOS Pathogens

Thomas Guillard

Section Editor

PLOS Pathogens

Michael Malim

Editor-in-Chief

PLOS Pathogens

orcid.org/0000-0002-7699-2064

Reviewer Comments (if any, and for reference):

Figure Files:

Data Requirements:

Reproducibility:

References:

---

## [Editor Report · Decision Letter 2]

1 Oct 2024

Dear Andy,

We are pleased to inform you that your manuscript 'A master regulator of central carbon metabolism directly activates virulence gene expression in attaching and effacing pathogens' has been provisionally accepted for publication in PLOS Pathogens.

Best regards,

John

John M Leong

Academic Editor

PLOS Pathogens

Thomas Guillard

Section Editor

PLOS Pathogens

Michael Malim

Editor-in-Chief

PLOS Pathogens

orcid.org/0000-0002-7699-2064
---

## [Editor Report · Acceptance letter]

5 Oct 2024

Dear Dr Roe,

We are delighted to inform you that your manuscript, "A master regulator of central carbon metabolism directly activates virulence gene expression in attaching and effacing pathogens," has been formally accepted for publication in PLOS Pathogens.

Best regards,

Michael Malim

Editor-in-Chief

PLOS Pathogens

orcid.org/0000-0002-7699-2064